# Transcriptional regulation of SARS-CoV-2 receptor ACE2 by SP1

Hui Han[1†], Rong-Hua Luo[2†], Xin-Yan Long[2†], Li-Qiong Wang[3†], Qian Zhu[1], Xin-Yue Tang[1], Rui Zhu[1], Yi-Cheng Ma[1]*, Yong-Tang Zheng[2]*, Cheng-Gang Zou[1]*

[1]State Key Laboratory for Conservation and Utilization of Bio-Resources in Yunnan, School of Life Sciences, Yunnan University, Kunming, China; [2]Key Laboratory of Bioactive Peptides of Yunnan Province/Key Laboratory of Animal Models and Human Disease Mechanisms of the Chinese Academy of Sciences, KIZ-CUHK Joint Laboratory of Bio-Resources and Molecular Research in Common Diseases, Center for Biosafety Mega-Science, Kunming Institute of Zoology, Chinese Academy of Sciences, Kunming, China; [3]Department of Pathology, Yan'an Hospital, Kunming Medical University, Kunming, China

**\*For correspondence:**
mayc@ynu.edu.cn (Y-CM);
zhengyt@mail.kiz.ac.cn (Y-TZ);
chgzou@ynu.edu.cn (C-GZ)

[†]These authors contributed equally to this work

**Competing interest:** The authors declare that no competing interests exist.

**Abstract** Angiotensin-converting enzyme 2 (ACE2) is a major cell entry receptor for severe acute respiratory syndrome coronavirus 2 (SARS-CoV-2). The induction of ACE2 expression may serve as a strategy by SARS-CoV-2 to facilitate its propagation. However, the regulatory mechanisms of ACE2 expression after viral infection remain largely unknown. Using 45 different luciferase reporters, the transcription factors SP1 and HNF4α were found to positively and negatively regulate ACE2 expression, respectively, at the transcriptional level in human lung epithelial cells (HPAEpiCs). SARS-CoV-2 infection increased the transcriptional activity of SP1 while inhibiting that of HNF4α. The PI3K/AKT signaling pathway, activated by SARS-CoV-2 infection, served as a crucial regulatory node, inducing ACE2 expression by enhancing SP1 phosphorylation—a marker of its activity—and reducing the nuclear localization of HNF4α. However, colchicine treatment inhibited the PI3K/AKT signaling pathway, thereby suppressing ACE2 expression. In Syrian hamsters (*Mesocricetus auratus*) infected with SARS-CoV-2, inhibition of SP1 by either mithramycin A or colchicine resulted in reduced viral replication and tissue injury. In summary, our study uncovers a novel function of SP1 in the regulation of ACE2 expression and identifies SP1 as a potential target to reduce SARS-CoV-2 infection.

## Editor's evaluation

This is a valuable report that describes that ACE2 expression is upregulated by SARS-CoV-2 infection via activation of transcription factor Sp1 and inhibition of HNF4α through the PI3K/AKT pathway. Inhibition of Sp1 reduces SARS-CoV-2 infection in vitro and in an animal model. This work is solid and will be of interest to those interested in ACE2 biology and its impact in COVID-19.

## Introduction

Severe acute respiratory syndrome coronavirus 2 (SARS-CoV-2) is the causative pathogen of the coronavirus disease 2019 (COVID-19) pandemic (*Wu et al., 2020*; *Lu et al., 2020*). Patients with severe COVID-19 often exhibit pathophysiological characteristics related to a systemic inflammatory response, manifesting as acute respiratory distress syndrome and multiorgan dysfunction (*Pan et al., 2021*; *van Eijk et al., 2021*). Viral entry into host cells occurs through the binding of the SARS-CoV-2 spike protein to its cellular receptor, angiotensin-converting enzyme 2 (ACE2), followed by proteolytic cleavage of the spike protein via the transmembrane serine protease TMPRSS2 (*Yan et al., 2020*),

facilitating the fusion of the viral and host cell membranes, a crucial step for viral internalization (*Hoffmann et al., 2020*). ACE2 is widely expressed in different human tissues, including the lungs, small intestine, kidneys, liver, testes, heart, and brain (*Dong et al., 2020*; *Verdecchia et al., 2020*). Such widespread expression of ACE2 may account for the multi-organ targeting observed with SARS-CoV-2.

Vaccination is regarded as an effective strategy for mitigating SARS-CoV-2 infections, as well as reducing hospitalization and mortality rates. Despite this, the evolving mutational landscape of SARS-CoV-2 has compromised the ability of certain individuals to generate a sufficient immune response through vaccination alone. Notably, the efficacy of existing vaccines has shown a marked decline against emergent variants such as Omicron (*Kuhlmann et al., 2022*; *Wilhelm et al., 2022*). Modifications in the spike protein of these variants have been observed to increase their binding affinity to ACE2, thereby enhancing transmissibility (*Liu et al., 2021*; *McCallum et al., 2022*). Consequently, need to identify and develop novel host-directed therapeutics against SARS-CoV-2 remains urgent, especially in vulnerable populations such as the elderly.

Accumulating evidence suggests that SARS-CoV-2 infection significantly up-regulates ACE2 expression (*Gao et al., 2022*; *Wei et al., 2021*; *Xu et al., 2021a*; *Zhuang et al., 2020*). Specifically, SARS-CoV-2 infection can increase the expression of HMGB1, which subsequently induces the expression of ACE2, possibly through an epigenetic mechanism (*Wei et al., 2021*). Furthermore, overexpression of the SARS-CoV-2 spike protein markedly up-regulates ACE2 expression, a process reliant on type I interferon signaling pathways, with ACE2 being identified as an interferon-stimulated gene (*Zhou et al., 2021*; *Ziegler et al., 2020*). Several studies have demonstrated that the androgen receptor positively regulates the expression of ACE2 at the transcriptional level (*Qiao et al., 2021*; *Samuel et al., 2020*). Importantly, attenuation of ACE2 expression via suppression of the androgen receptor signaling decreases SARS-CoV-2 infectivity (*Qiao et al., 2021*). Recent research has also demonstrated that ursodeoxycholic acid (UDCA), an inhibitor of the farnesoid X receptor (FXR), reduces ACE2 expression in human lung, intestinal, and liver organoids, thereby inhibiting SARS-CoV-2 infection (*Brevini et al., 2023*). Accordingly, ACE2 represents a promising therapeutic target in combating COVID-19, although the molecular mechanisms related to SARS-CoV-2-induced regulation of ACE2 remain largely unknown.

To elucidate the molecular mechanisms governing the regulation of ACE2 expression by SARS-CoV-2, we employed 45 different luciferase reporters to assay a range of signaling pathways. Our results indicated that SARS-CoV-2 up-regulated ACE2 expression by activating the transcription factor SP1, while concurrently inhibiting HNF4α via the PI3K/AKT signaling pathway. Furthermore, we demonstrated that mithramycin A (MithA), an inhibitor of SP1, exhibited efficacy against SARS-CoV-2 in both cellular and animal models. Thus, these findings suggest that SP1 serves as an important transcription factor in the regulation of ACE2 expression.

## Results

### SARS-CoV-2 infection up-regulates ACE2 expression, which is inhibited by colchicine treatment

Serving as the entry receptor for SARS-CoV-2 in host cells, ACE2 is considered a promising therapeutic target against COVID-19 (*Monteil et al., 2020*). Consistent with prior studies indicating a significant up-regulation of ACE2 mRNA expression following SARS-CoV-2 infection (*Gao et al., 2022*; *Wei et al., 2021*; *Xu et al., 2021b*; *Zhuang et al., 2020*), we found that SARS-CoV-2 infection up-regulated ACE2 protein expression in HPAEpiCs, a human lung epithelial cell line (*Figure 1A, B*), with further corroboration based on immunofluorescence analysis (*Figure 1C, D*). Recent clinical investigations have reported a mortality benefit among COVID-19 patients treated with colchicine (*Drosos et al., 2022*; *Elshafei et al., 2021*), a drug used for the treatment of (auto-)inflammatory conditions such as acute gout and familial Mediterranean fever (*Dasgeb et al., 2018*; *Schlesinger et al., 2020*). In the present study, we observed that colchicine treatment substantially inhibited ACE2 expression in HPAEpiCs, irrespective of SARS-CoV-2 infection status (*Figure 1A–D*).

As colchicine inhibits ACE2 expression, we assessed the in vitro inhibitory effects of colchicine on SARS-CoV-2 replication in HPAEpiCs. After preincubation with colchicine at different concentrations for 1 hr, cells were infected with SARS-CoV-2 for 1 hr, then cultured in fresh medium for 24 hr to measure viral RNA copy number. Based on quantitative reverse transcriptase polymerase

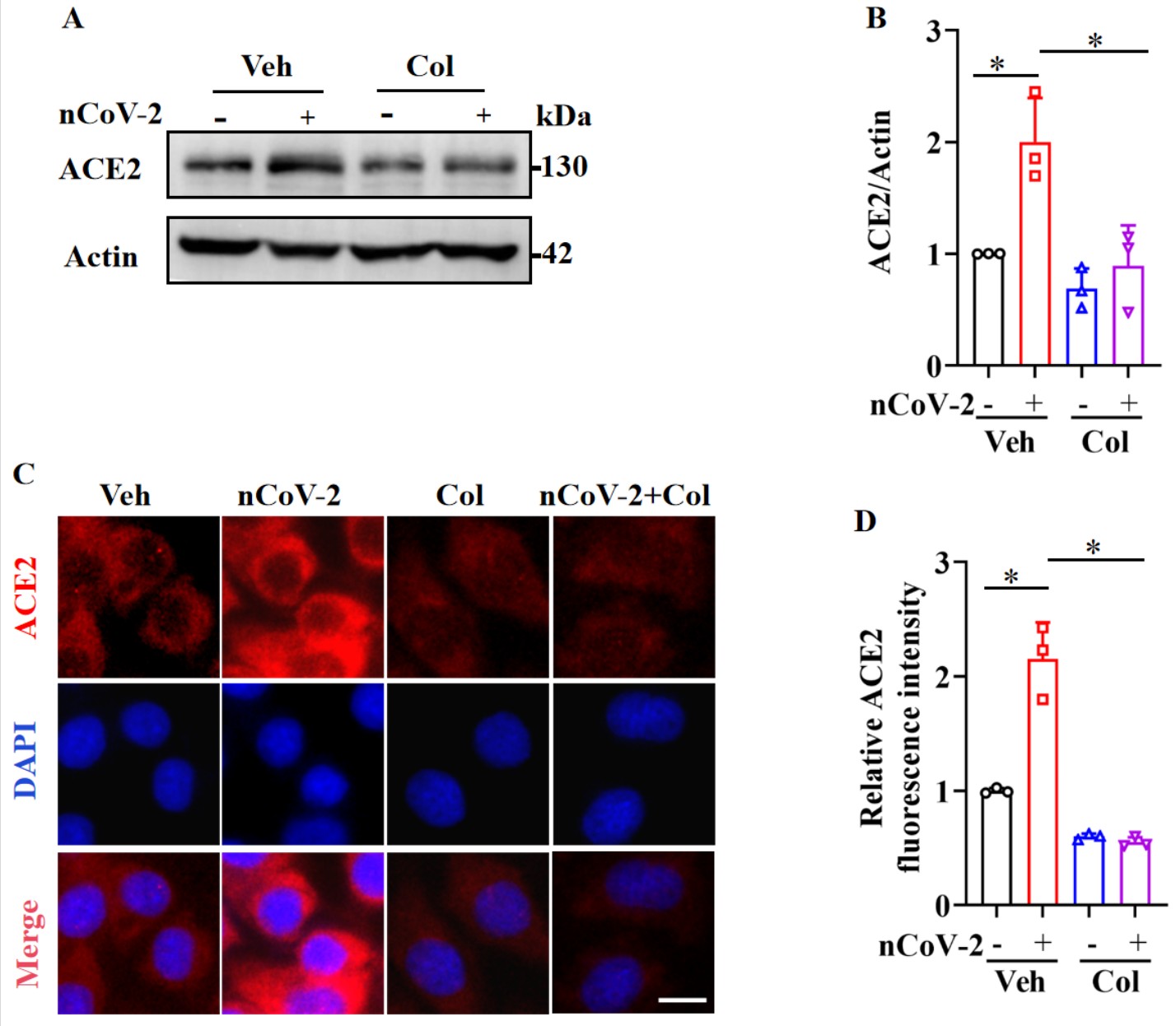

**Figure 1.** SARS-CoV-2 infection up-regulated ACE2 expression, which was suppressed by colchicine. (**A** and **B**) SARS-CoV-2 infection up-regulated protein levels of ACE2. Colchicine (20 nM) significantly reduced protein levels of ACE2 in HPAEpiCs. Blot is typical of three independent experiments (**A**). Quantification of ACE2 to Actin ratio (**B**). Results are means ± standard deviation (SD) of three independent experiments. *p<0.05 (Student's *t*-test). (**C**) Representative images of immunofluorescence staining for ACE2. Scale bar: 10 µm. (**D**) Quantification of ACE2 fluorescence intensity. Results are means ± SD of three independent experiments. *p<0.05 (Student's *t*-test). Veh, Vehicle. Col, Colchicine. nCoV-2, SARS-CoV-2.

The online version of this article includes the following source data and figure supplement(s) for figure 1:

**Source data 1.** Original uncropped western blot images in *Figure 1A* (anti-ACE2 and anti-Actin).

**Source data 2.** PDF containing *Figure 1A* and original scans of relevant western blots (anti-ACE2 and anti-Actin) with highlighted bands and sample labels.

**Source data 3.** Original file for quantification of ACE2 to Actin ratio in *Figure 1B* (anti-ACE2 and anti-Actin).

**Source data 4.** Original file for quantification of ACE2 fluorescence intensity in *Figure 1D*.

**Figure supplement 1.** Colchicine blocked SARS-CoV-2 replication.

**Figure supplement 1—source data 1.** Original file for determination of viral load in *Figure 1—figure supplement 1A*.

**Figure supplement 1—source data 2.** Original file for dose-response analysis in *Figure 1—figure supplement 1B*.

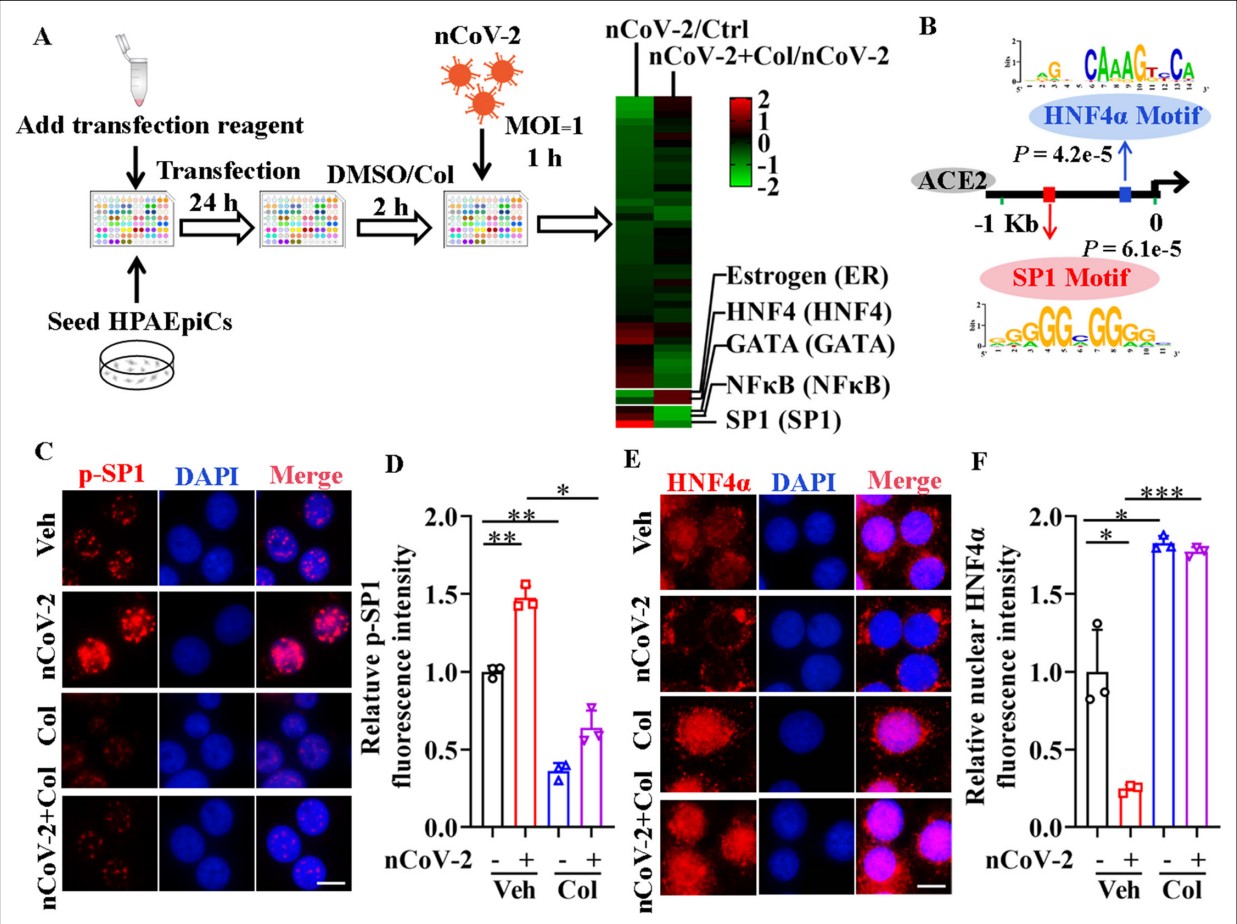

**Figure 2.** Activation of SP1 and inactivation of HNF4α were mediated by SARS-CoV-2. (**A**) Assays for signaling pathways in response to SARS-CoV-2 infection and colchicine treatment (20 nM) using Cignal Finder 45-Pathway Reporter Array. (**B**) SP1 and HNF4α binding elements 1.5 kb upstream of TSS of ACE2 gene identified by MEME program. (**C** and **D**) SARS-CoV-2 significantly increased phosphorylation of SP1 (p–SP1) in HPAEpiCs, which was suppressed by treatment with colchicine (20 nM). Representative images of immunofluorescence staining for p-SP1 (**C**). Scale bar: 10 µm. Quantification of p-SP1 fluorescence intensity (**D**). Results are means ± SD of three independent experiments. *p<0.05, **p<0.01 (Student's t-test). (**E** and **F**) SARS-CoV-2 induced cytoplasmic translocation of HNF4α, whereas colchicine (20 nM) promoted its nuclear accumulation in HPAEpiCs. Representative images of immunofluorescence staining for HNF4α (**E**). Scale bar: 10 µm. Quantification of HNF4α fluorescence intensity (**F**). Results are means ± SD of three independent experiments. *p<0.05, ***p<0.001 (Student's t-test). Veh, Vehicle. Col, Colchicine. nCoV-2, SARS-CoV-2.

The online version of this article includes the following source data and figure supplement(s) for figure 2:

**Source data 1.** Original file for quantification of p-SP1 fluorescence intensity in *Figure 2D*.

**Source data 2.** Original file for quantification of HNF4α fluorescence intensity in *Figure 2F*.

**Figure supplement 1.** SP1 was primarily located in nucleus of HPAEpiCs.

chain reaction (RT-qPCR), colchicine treatment reduced extracellular SARS-CoV-2 replication in cells (*Figure 1—figure supplement 1A*). The half-maximal effective concentration ($EC_{50}$) of colchicine for inhibiting viral replication was 0.2703 µM (*Figure 1—figure supplement 1B*). These results suggest that inhibition of ACE2 expression by colchicine suppresses SARS-CoV-2 infection.

## SARS-CoV-2 infection regulates transcriptional activities of SP1 and HNF4α

To clarify the molecular mechanisms governing ACE2 expression regulation, we used colchicine as a proof-of-principle agent. Using the Cignal Finder 45-Pathway Reporter Array (*Manzini et al., 2014*; *Xu et al., 2021a*), we analyzed a range of signaling pathways in response to SARS-CoV-2 infection, both in the presence and absence of colchicine (*Figure 2A*). Among the tested signaling pathways, three transcription factors (SP1, NF-κB, and GATA) enhanced by SARS-CoV-2 infection and suppressed by

colchicine treatment (*Figure 2A*). Conversely, two signaling pathways (HNF4α and estrogen receptor) were inactivated by SARS-CoV-2 infection but activated by colchicine treatment (*Figure 2A*). These transcription factors thus emerged as potential candidates for regulating ACE2 expression. To further explore this, we investigated the DNA motifs located 1.5 kb upstream of the transcription start site (TSS) of the ACE2 gene using the MEME program (*Bailey et al., 2015*). Our analysis revealed two significantly enriched motifs (*Figure 2B*), annotated as the motifs of the transcription factors SP1 (p=6.1e-5) and HNF4α (p=4.2e-5). These findings suggest that these two transcription factors are likely regulators of ACE2 expression.

To assess the impact of SARS-CoV-2 and colchicine on SP1 and HNF4α activities, we examined their subcellular distributions using immunofluorescence analysis. Although SP1 was mainly located in the nucleus of HPAEpiCs (*Figure 2—figure supplement 1*), only a small proportion of total SP1 was phosphorylated at Thr453 (*Figure 2C*), a modification indicative of its activation (*Milanini-Mongiat et al., 2002*). SARS-CoV-2 infection markedly increased the phosphorylation levels of SP1 at Thr453, whereas colchicine treatment significantly inhibited the phosphorylation of SP1, regardless of the presence or absence of SARS-CoV-2 infection (*Figure 2C, D*). In contrast to SP1, HNF4α displayed both nuclear and cytoplasmic distribution in HPAEpiCs under basal conditions (*Figure 2E*). SARS-CoV-2 infection prompted a redistribution of HNF4α from the nucleus to cytoplasm, while colchicine treatment induced its nuclear accumulation both in the presence and absence of SARS-CoV-2 infection (*Figure 2E, F*). These results suggest that SARS-CoV-2 infection activates SP1 and inactivates HNF4α, which can be counteracted by colchicine treatment.

## Involvement of SP1 and HNF4α in ACE2 expression

Next, we investigated whether colchicine inhibited ACE2 expression via regulation of SP1 and HNF4α. First, western blot analysis demonstrated that treatment with MithA, a selective SP1 inhibitor, or knockdown of SP1 by small interfering RNA (siRNA) down-regulated ACE2 protein expression in HPAEpiCs (*Figure 3A, B*). However, inhibition of SP1 by MithA or siSP1 did not further reduce the down-regulation of ACE2 protein levels achieved by colchicine. In contrast, treatment with the HNF4α antagonist BI6015 and knockdown of HNF4α by siRNA up-regulated ACE2 protein expression (*Figure 3C, D*), which was blocked by colchicine administration.

Immunofluorescence staining further confirmed that although MithA treatment inhibited the protein expression of ACE2, it did not further reduce the expression of ACE2 suppressed by colchicine (*Figure 3E, F*). Supplementation with BI6015 markedly increased the expression of ACE2 in HPAEpiCs, which was reduced by colchicine treatment (*Figure 3E, F*). Similar results were obtained in human epithelial cells A549, human renal tubular cells HK-2, and human hepatoma cells Huh-7 (*Figure 3—figure supplements 1 and 2*).

Based on a luciferase reporter gene containing the ACE2 promoter, MithA treatment inhibited luciferase activity, but not in the presence of colchicine (*Figure 3G*), while BI6015 treatment increased luciferase activity. In addition, chromatin immunoprecipitation (ChIP)-qPCR analysis indicated that the binding of SP1 to the GC box of the ACE2 promoter was significantly reduced, while the binding of HNF4α to the AGGTCA element was markedly increased after colchicine treatment (*Figure 3H*). As MithA inhibited the expression of ACE2, we also tested its effects on SARS-CoV-2 infection in vitro. Results showed that MithA inhibited SARS-CoV-2 replication in HPAEpiCs, with an $EC_{50}$ value of 0.1948 μM (*Figure 3—figure supplement 3*). These data suggest that SP1 and HNF4α exert opposing effects on the transcriptional regulation of ACE2 expression, thereby influencing cellular susceptibility to SARS-CoV-2 infection.

## Dual antagonism of SP1 and HNF4α

Interestingly, treatment with the SP1 inhibitor MithA markedly suppressed SP1 phosphorylation, whereas treatment with the HNF4α antagonist BI6015 increased SP1 phosphorylation at Thr453 (*Figure 4A, B*). In addition, BI6015 induced the cytoplasmic translocation of HNF4α, while MithA promoted its nuclear accumulation (*Figure 4C, D*). Western blot analysis showed an elevation in SP1 phosphorylation levels following the knockdown of HNF4α by siRNA (*Figure 4E, F*), whereas the knockdown of SP1 had no discernible impact on the total levels of HNF4α. We subsequently confirmed potential interactions between SP1 and HNF4α in HPAEpiCs using co-immunoprecipitation

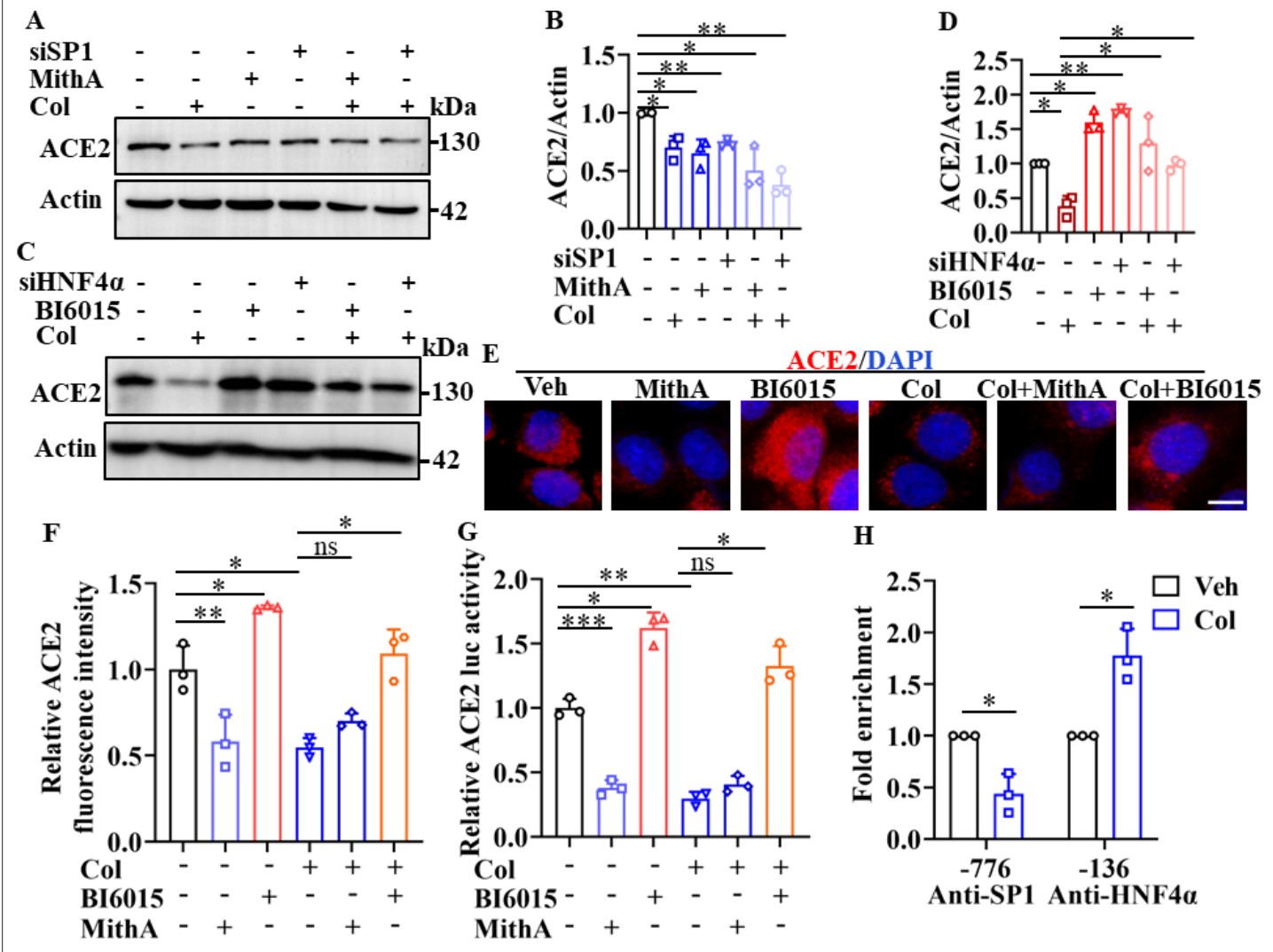

**Figure 3.** SP1 and HNF4α exerted opposing effects on regulation of ACE2 expression. (**A** and **B**) HPAEpiCs were treated with colchicine (20 nM), MithA (100 nM), colchicine +MithA, siSP1, or siSP1 +colchicine, respectively. Inhibition of SP1 significantly suppressed ACE2 levels in HPAEpiCs. Blot is typical of three independent experiments (**A**). Quantification of ACE2 to Actin ratio (**B**). Results are means ± SD of three independent experiments. *p<0.05, **p<0.01 (Student's *t*-test). (**C** and **D**) Supplementation with colchicine (20 nM) significantly suppressed ACE2 levels in HPAEpiCs, which was reversed by treatment with BI6015 (20 μM) or siHNF4α. Blot is typical of three independent experiments (**C**). Quantification of ACE2 to Actin ratio (**D**). Results are means ± SD of three independent experiments. *p<0.05, **p<0.01 (Student's *t*-test). (**E** and **F**) HPAEpiCs were treated with MithA (100 nM), BI6015 (20 μM), colchicine (20 nM), colchicine + MithA, or colchicine +BI6015, respectively. Representative images of immunofluorescence staining for ACE2 in HPAEpiCs (**E**). Scale bar: 10 μm. Quantification of ACE2 fluorescence intensity (**F**). Results are means ± SD of three independent experiments. *p<0.05, **p<0.01, ns, not significant (Student's *t*-test). (**G**) Luciferase activity analysis of ACE2 promoter in HPAEpiCs. Results are means ± SD of three independent experiments. *p<0.05, **p<0.01, ***p<0.001, ns, not significant (Student's *t*-test). (**H**) Putative SP1 and HNF4α binding sites in promoter regions of ACE2 were detected by ChIP-qPCR with anti-SP1 and anti-HNF4α antibodies, respectively. Results are means ± SD of three independent experiments. *p<0.05. Veh, Vehicle. Col, Colchicine.

The online version of this article includes the following source data and figure supplement(s) for figure 3:

**Source data 1.** Original uncropped western blot images in *Figure 3A* (anti-ACE2 and anti-Actin).

**Source data 2.** Original uncropped western blot images in *Figure 3C* (anti-ACE2 and anti-Actin).

**Source data 3.** PDF containing *Figure 3A, C* and original scans of relevant western blot analysis (anti-ACE2 and anti-Actin) with highlighted bands and sample labels.

**Source data 4.** Original file for quantification of ACE2 to Actin ratio in *Figure 3B, D* (anti-ACE2 and anti-Actin).

**Source data 5.** Original file for quantification of ACE2 fluorescence intensity in *Figure 3F*.

**Source data 6.** Original file for luciferase activity analysis in *Figure 3G*.

*Figure 3 continued on next page*

*Figure 3 continued*

**Source data 7.** Original file for ChIP analysis in *Figure 3H*.

**Figure supplement 1.** Immunofluorescence analysis of expression and localization of ACE2 in A549 cells.

**Figure supplement 1—source data 1.** Original file for quantification of ACE2 fluorescence intensity in *Figure 3—figure supplement 1B*.

**Figure supplement 2.** SARS-CoV-2 infection up-regulated ACE2 expression, which was suppressed by colchicine in HK-2 and Huh-7 cells.

**Figure supplement 2—source data 1.** Original uncropped western blot images in *Figure 3—figure supplement 2A* (anti-ACE2 and anti-Actin).

**Figure supplement 2—source data 2.** Original uncropped western blot images in *Figure 3—figure supplement 2C* (anti-ACE2 and anti-Actin).

**Figure supplement 2—source data 3.** PDF containing *Figure 3—figure supplement 2A, C* and original scans of relevant western blot analysis (anti-ACE2 and anti-Actin) with highlighted bands and sample labels.

**Figure supplement 2—source data 4.** Original file for quantification of ACE2 to Actin ratio in *Figure 3—figure supplement 2B, D* (anti-ACE2 and anti-Actin).

**Figure supplement 2—source data 5.** Original file for determination of viral load in *Figure 3—figure supplement 2E, F*.

**Figure supplement 3.** MithA inhibited SARS-CoV-2 replication in a dose-dependent manner.

**Figure supplement 3—source data 1.** Original file for dose-response analysis in *Figure 3—figure supplement 3*.

assay (co-IP; *Figure 4G*). Overall, these findings suggest that SP1 and HNF4α antagonize each other, likely through protein-protein interactions.

## Colchicine reduces ACE2 expression by inhibiting the PI3K/AKT signaling pathway

The PI3K/AKT signaling pathway is reportedly activated by SARS-CoV-2 infection (*Callahan et al., 2021*; *Klann et al., 2020*; *Sun et al., 2021*). Consistently, we found that SARS-CoV-2 infection enhanced the phosphorylation of AKT at Ser473 and Thr308, which is required for its activation (*Su et al., 2011*; *Figure 5A–C*). However, treatment with colchicine markedly reduced the phosphorylation of AKT induced by SARS-CoV-2 infection (*Figure 5A–C*). Furthermore, knockdown of AKT by siRNA or treatment with PI3K inhibitors (LY294002 and wortmannin) down-regulated the expression of ACE2 in HPAEpiCs (*Figure 5D–G*). Notably, both LY294002 and wortmannin inhibited SARS-CoV-2 infection in HPAEpiCs, with $EC_{50}$ values of 0.2381 μM and 0.04228 μM, respectively (*Figure 5—figure supplement 1*). These data suggest that the PI3K/AKT signaling pathway is essential for effective SARS-CoV-2 infection.

The PI3K/AKT signaling pathway plays an important role in regulating the transcriptional activities of SP1 and HNF4α (*Adapala et al., 2019*; *Gómez-Villafuertes et al., 2015*; *Li et al., 2019*; *Zhao et al., 2015*). Activation of AKT promotes the stability and localization of SP1 by phosphorylating SP1 at Thr453 and Thr739 (*Adapala et al., 2019*; *Gómez-Villafuertes et al., 2015*; *Zhao et al., 2015*), and prevents the nuclear translocation of HNF4α (*Li et al., 2019*). Consistent with these observations, we found that suppression of AKT by siRNA or its inhibitors significantly reduced the accumulation of phospho-SP1 in the nucleus, but markedly promoted the nuclear translocation of HNF4α (*Figure 5H–K*). Taken together, these findings suggest that the down-regulation of ACE2 expression by colchicine is contingent upon the inhibition of the PI3K/AKT signaling pathway, which, in turn, modulates the transcriptional activities of SP1 and HNF4α.

## Inhibition of SP1 reduces viral load and damage to the respiratory and renal systems

We tested whether inhibition of SP1 by colchicine and MithA inhibits the replication of SARS-CoV-2 in vivo using Syrian hamsters (*Mesocricetus auratus*), an animal model used for the study of COVID-19 pneumonia and therapeutic evaluation (*Chan et al., 2020*; *Choudhary et al., 2022*; *Muñoz-Fontela et al., 2020*; *Rosenke et al., 2021*; *Sia et al., 2020*). These hamsters were intranasally infected with SARS-CoV-2 at a 50% tissue culture infectious dose ($TCID_{50}$) of $10^4$. After 1 hr, the hamsters were inoculated intraperitoneally with colchicine or MithA at 0.2 mg/kg, respectively. A mock group was treated with vehicle (1% DMSO and 99% saline) using the same route and timing (*Figure 6—figure supplement 1*). The animals were dosed every 24 hr with either colchicine or MithA at 0.2 mg/kg, respectively. Lung and tracheal samples were collected at 3 days post-infection (dpi) to assess viral RNA and ACE2 expression. Immunofluorescence analysis demonstrated that the expression of ACE2 was

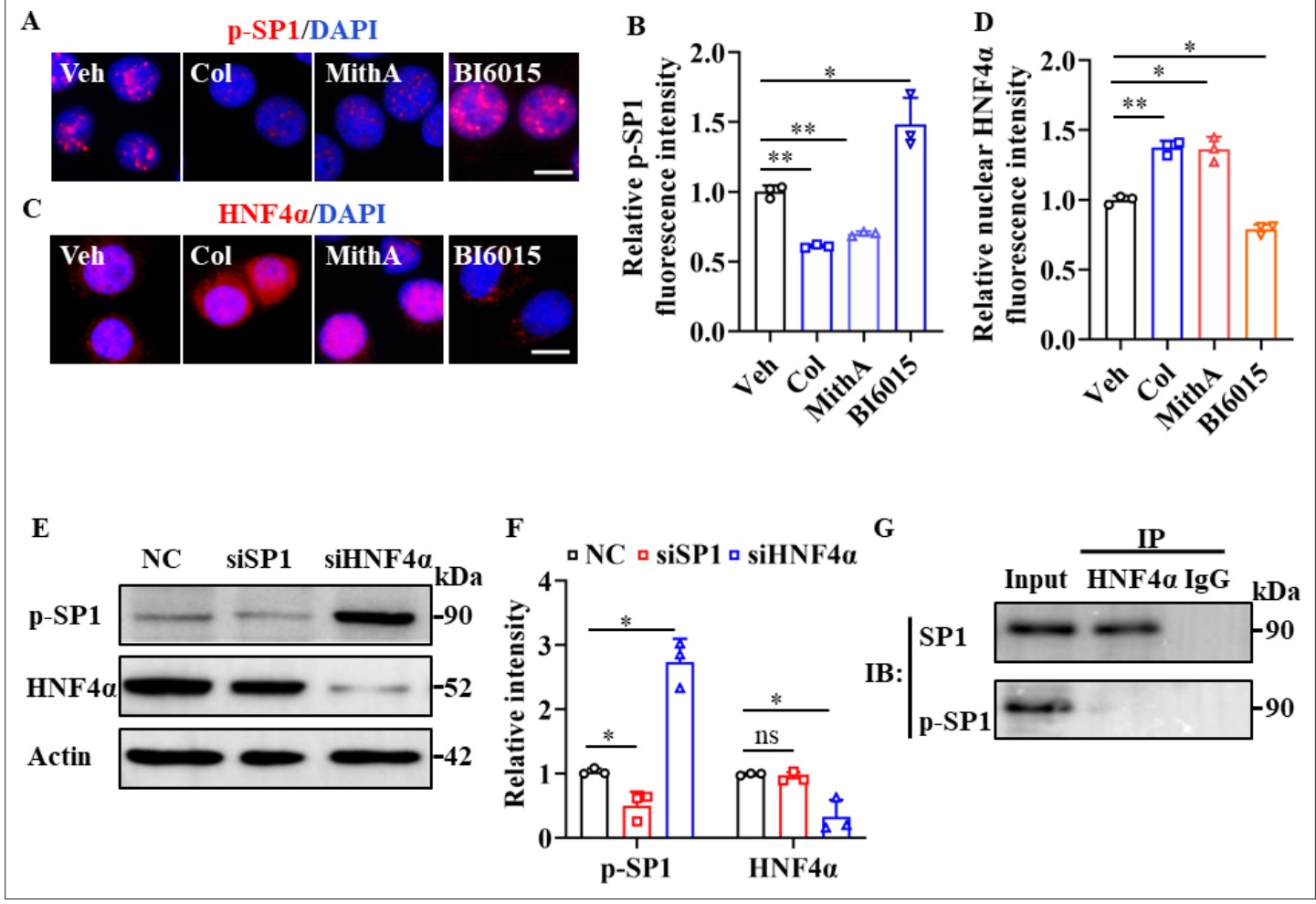

**Figure 4.** SP1 and HNF4α antagonized each other via protein-protein interactions. (**A** and **B**) Supplementation with colchicine (20 nM) and MithA (100 nM) significantly suppressed SP1 phosphorylation (p–SP1) in HPAEpiCs, which was reversed by treatment with BI6015 (20 µM). Representative images of immunofluorescence staining for p-SP1 (**A**). Scale bar: 10 µm. Quantification of p-SP1 fluorescence intensity (**B**). Results are means ± SD of three independent experiments. *p<0.05, **p<0.01 (Student's *t*-test). (**C** and **D**) Supplementation with colchicine (20 nM) and MithA (100 nM) promoted nuclear accumulation of HNF4α in HPAEpiCs, which was inhibited by treatment with BI6015 (20 µM). Representative images of immunofluorescence staining for HNF4α (**C**). Scale bar: 10 µm. Quantification of HNF4α fluorescence intensity (**D**). Results are means ± SD of three independent experiments. *p<0.05, **p<0.01 (Student's *t*-test). (**E** and **F**) Phosphorylation levels of SP1 and total protein levels of HNF4α were measured in HPAEpiCs by western blotting. Blot is typical of three independent experiments (**E**). Quantification of p-SP1 or HNF4α to Actin ratio (**F**). Results are means ± SD of three independent experiments. *p<0.05, ns, not significant (Student's *t*-test). (**G**) Interactions between SP1 and HNF4α were measured by co-IP in HPAEpiCs. NC, Negative control. Veh, Vehicle. Col, Colchicine.

The online version of this article includes the following source data for figure 4:

**Source data 1.** Original file for quantification of p-SP1 fluorescence intensity in *Figure 4B*.

**Source data 2.** Original file for quantification of HNF4α fluorescence intensity in *Figure 4D*.

**Source data 3.** Original uncropped western blot images in *Figure 4E* (anti-p-SP1, anti-HNF4α, and anti-Actin).

**Source data 4.** PDF containing *Figure 4E, G* and original scans of relevant western blot analysis (anti-p-SP1, anti-HNF4α, and anti-Actin) with highlighted bands and sample labels.

**Source data 5.** Original file for quantification of p-SP1 or HNF4α to Actin ratio in *Figure 4F* (anti-p-SP1, anti-HNF4α, and anti-Actin).

**Source data 6.** Original uncropped western blot images in *Figure 4G*.

significantly up-regulated in the lungs and trachea of hamsters infected with SARS-CoV-2 compared to the non-infected control group (*Figure 6A*; *Figure 6—figure supplement 2A, B*). However, colchicine or MithA treatment substantially inhibited the expression of ACE2 in both the lungs and trachea of hamsters infected with SARS-CoV-2. Based on immunofluorescence analysis of SARS-CoV-2

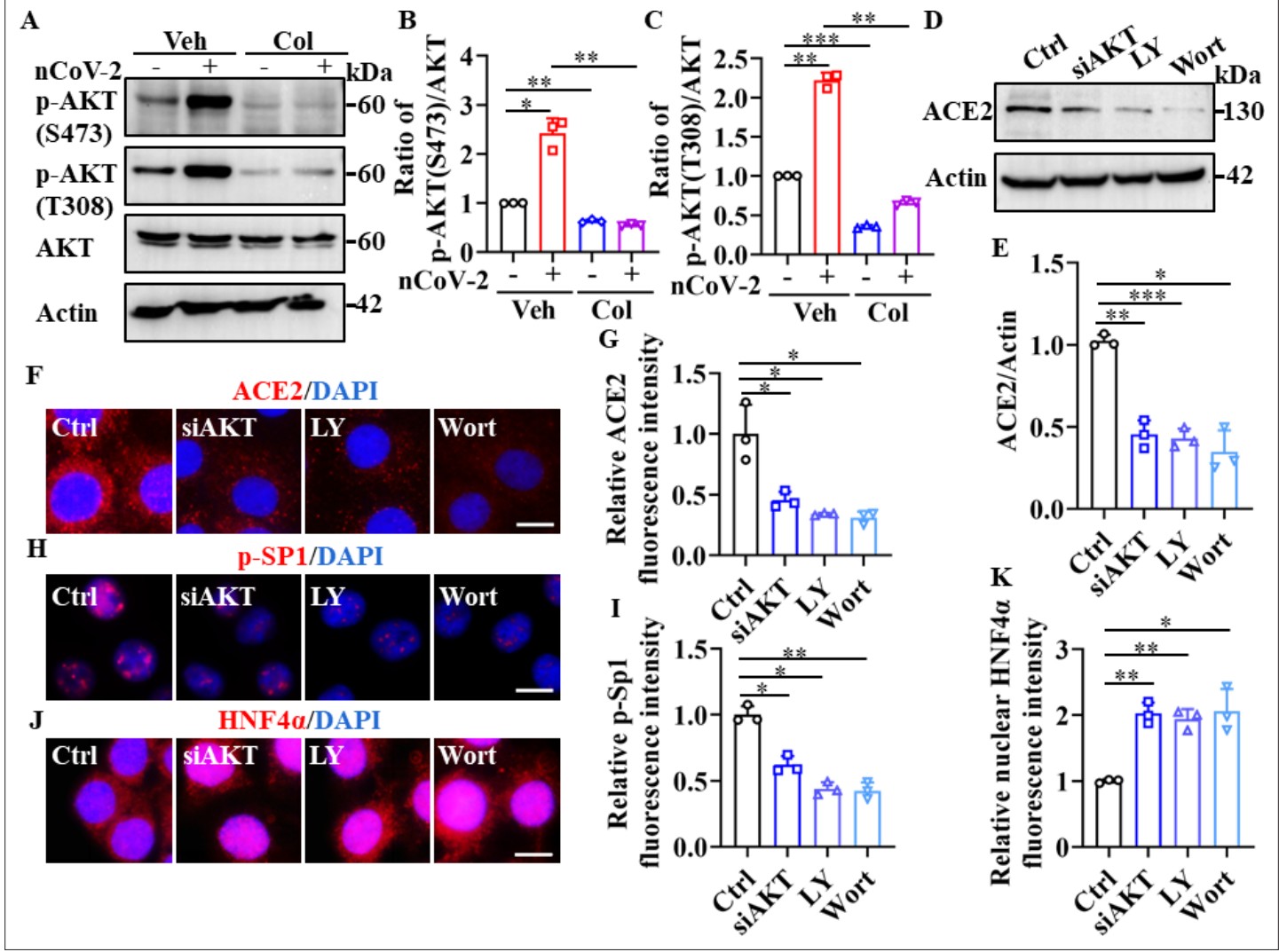

**Figure 5.** SP1 and HNF4α were downstream of the PI3K/AKT signaling pathway. (**A–C**) Phosphorylation levels of AKT (S473 and T308) were measured in HPAEpiCs by western blotting. Blot is typical of three independent experiments (**A**). Quantification of p-AKT (S473) (**B**) or p-AKT (T308) (**C**) to Actin ratio. Results are means ± SD of three independent experiments. *p<0.05, **p<0.01, ***p<0.001 (Student's *t*-test). (**D–G**) Supplementation with LY294002 (30 µM) and wortmannin (50 nM) or siAKT significantly suppressed ACE2 levels in HPAEpiCs. Blot is typical of three independent experiments (**D**). Quantification of ACE2 to Actin ratio (**E**). Results are means ± SD of three independent experiments. *p<0.05, **p<0.01, ***p<0.001 (Student's *t*-test). Representative images of immunofluorescence staining for ACE2 (**F**). Scale bar: 10 µm. Quantification of ACE2 fluorescence intensity (**G**). Results are means ± SD of three independent experiments. *p<0.05 (Student's *t*-test). (**H** and **I**) Supplementation with LY294002 (30 µM) and wortmannin (50 nM) or siAKT significantly suppressed phosphorylation of SP1 (p–SP1) in HPAEpiCs. Representative images of immunofluorescence staining for p-SP1 (**H**). Scale bar: 10 µm. Quantification of p-SP1 fluorescence intensity (**I**). Results are means ± SD of three independent experiments. **p<0.01, **p<0.01 (Student's *t*-test). (**J** and **K**) Supplementation with LY294002 (30 µM) and wortmannin (50 nM) or siAKT promoted nuclear accumulation of HNF4α in HPAEpiCs. Representative images of immunofluorescence staining for HNF4α (**J**). Scale bar: 10 µm. Quantification of HNF4α fluorescence intensity (**K**). Results are means ± SD of three independent experiments. *p<0.05, **p<0.01 (Student's *t*-test). Ctrl, Control. Veh, Vehicle. Col, Colchicine. LY, LY294002. Wort, wortmannin. nCoV-2, SARS-CoV-2.

The online version of this article includes the following source data and figure supplement(s) for figure 5:

**Source data 1.** Original uncropped western blot images in **Figure 5A** (anti-p-AKT (S473), anti-p-AKT (T308), and anti-Actin).

**Source data 2.** PDF containing **Figure 5A, D** and original scans of relevant western blot analysis with highlighted bands and sample labels.

**Source data 3.** Original file for quantification of p-AKT to AKT ratio in **Figure 5B, C**.

**Source data 4.** Original uncropped western blot images in **Figure 5D** (anti-ACE2 and anti-Actin).

**Source data 5.** Original file for quantification the ratio of ACE2 to Actin in **Figure 5E**.

**Source data 6.** Original files for quantification of ACE2, p-SP1, and HNF4α fluorescence intensity in **Figure 5G, I and K**.

*Figure 5 continued on next page*

*Figure 5 continued*

**Figure supplement 1.** PI3K inhibitors inhibited SARS-CoV-2 replication in a dose-dependent manner.

**Figure supplement 1—source data 1.** Original file for dose-response analysis in *Figure 5—figure supplement 1A, B*.

nucleocapsid and RT-qPCR, treatment with colchicine or MithA reduced viral replication in both the lungs and trachea of hamsters (*Figure 6B–D*; *Figure 6—figure supplement 2C, D*), respectively.

Subsequent histopathological analysis of lung tissues 3 dpi revealed a range of pulmonary abnormalities, including bronchial epithelial cell necrosis and nuclear pyknosis, extensive alveolar hemorrhage, marked infiltration of inflammatory cells, significant edema, and notably thickened alveolar walls (*Figure 6E, G*). Although similar pathological features were present, their severity was markedly reduced in the treatment groups compared to the mock group. Using Masson's trichrome staining, collagen deposition indicative of fibrosis was evident in the lungs of infected hamsters (*Figure 6F, H*). Of note, treatment with either colchicine or MithA effectively reversed lung fibrosis. These findings suggest that colchicine and MithA antagonize SARS-CoV-2 replication in the lung and trachea while also ameliorating associated histopathological damage.

Although acute kidney injury associated with severe SARS-CoV-2 serves as an independent risk factor for in-hospital death in patients (*Nadim et al., 2020*), whether SARS-CoV-2 directly infects the kidney remains unclear (*Smith and Akilesh, 2021*; *Wysocki et al., 2021*). Prevailing evidence appears to favor indirect kidney injury by SARS-CoV-2 (*Smith and Akilesh, 2021*). Here, immunofluorescence analysis detected the substantial presence of SARS-CoV-2 in the kidneys of infected hamsters, an effect markedly mitigated by colchicine or MithA treatment (*Figure 7A, B*). Further analysis also showed that both drugs effectively down-regulated ACE2 expression (*Figure 7C, D*). Histopathological evaluation showed significant renal damage in SARS-CoV-2-infected hamsters, including renal tubular epithelial cell nuclear pyknosis, brush border disappearance, renal interstitial vascular congestion, inflammatory cell infiltration, and glomerular atrophy (*Figure 7E, F*). In the drug-treated hamsters, however, the glomeruli displayed uniform distribution and intact structure, tubular epithelial cells appeared normal, and brush borders showed neat arrangement, with no evident abnormality in the renal medulla or obvious hyperplasia in the renal interstitium. Collectively, these findings underscore the therapeutic efficacy of colchicine and MithA in ameliorating renal histopathological alterations.

## Discussion

Based on our study and current understanding of SARS-CoV-2 infection, a mechanistic model is proposed regarding how SARS-CoV-2 infection induces ACE2 expression via two transcription factors, SP1 and HNF4α, in host cells. Under physiological conditions, SP1 is primarily localized in the nucleus, while HNF4α exhibits a dual distribution in the nucleus and cytoplasm. These transcription factors exert opposing effects on the regulation of ACE2 expression, thereby maintaining basal expression of ACE2. Upon infection by SARS-CoV-2, the PI3K/AKT signaling pathway becomes activated (*Klann et al., 2020*). This activation subsequently promotes the transcriptional activity of SP1 through increased phosphorylation in the nucleus and suppresses the transcriptional activity of HNF4α by inducing its translocation to the cytoplasm. Disruption of this balance leads to the up-regulation of ACE2 after viral infection. Given that ACE2 serves as the entry receptor for SARS-CoV-2, induction of ACE2 expression may represent a strategic adaptation of the virus to facilitate its own propagation.

Recent advances in multi-omics approaches have facilitated our understanding of host responses to SARS-CoV-2, thereby accelerating the development and repositioning of therapeutic agents against COVID-19 (*Chu et al., 2021*; *Ho et al., 2021*; *Kamel et al., 2021*; *Klann et al., 2020*; *Lu et al., 2022*). Phosphoproteomic analysis of SARS-CoV-2-infected cells has revealed the activation of the growth factor receptor (GFR) and its downstream pathways, including the RAF/MEK/ERK MAPK and PI3K/AKT/mTOR signaling pathways, in SARS-CoV-2-infected cells (*Klann et al., 2020*). Blockage of the GFR signaling pathway via application of prominent anti-cancer drugs, such as sorafenib (RAF inhibitor), RO5126766 (dual RAF/MEK inhibitor), pictilisib (PI3K inhibitor), and omipalisib (dual PI3K and mTOR inhibitor), effectively prevents SARS-CoV-2 replication in cellular models. Here, through screening 45 signaling pathways in SARS-CoV-2-infected HPAEpiCs, either in the presence or absence of colchicine, our study elucidated the opposite regulatory roles of two transcription factors, SP1 and HNF4α, in the modulation of ACE2 expression at the transcriptional level, which exerted positive and

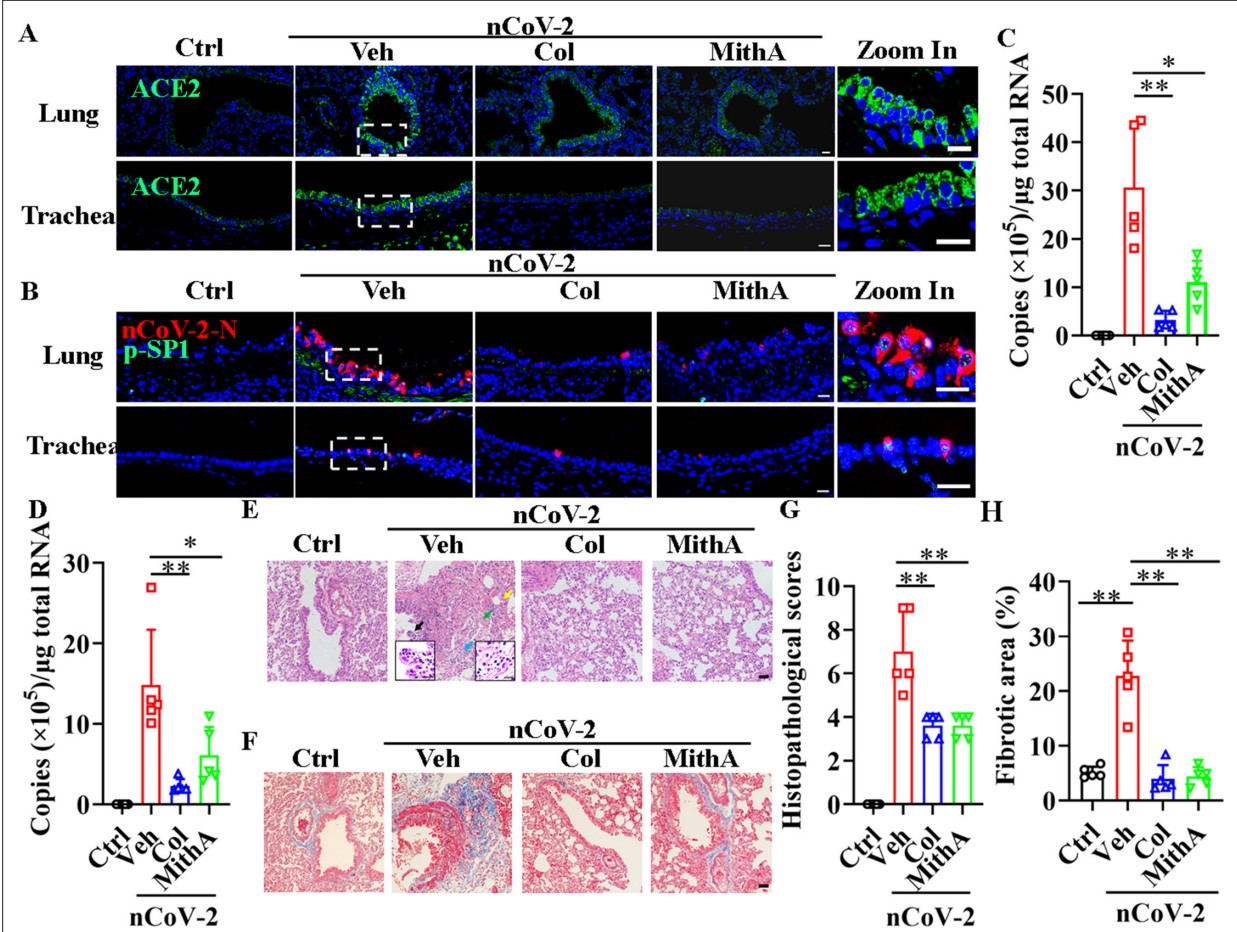

**Figure 6.** Inhibition of SP1 inhibited replication of SARS-CoV-2 and reduced lung pathology in Syrian hamsters. (**A**) Treatment with colchicine or MithA inhibited ACE2 expression in lung and trachea of hamsters infected with SARS-CoV-2. Representative images of immunofluorescence staining of ACE2 in lung and trachea of hamsters. Parts on right side are high-power images. Scale bar: 20 μm. (**B**) Treatment with colchicine or MithA inhibited replication of SARS-CoV-2 in lung and trachea of hamsters. Representative images of immunofluorescence staining of SARS-CoV-2-N and p-SP1 in lung and trachea of hamsters. Parts on right side are high-power images. Scale bar: 20 μm. Viral loads of SARS-CoV-2 were measured in lung (**C**) and trachea (**D**) of hamsters by RT-qPCR (n=5 each group). Error bars show means ± SD. *p<0.05, **p<0.01 (Student's t-test). (**E–H**) Supplementation with colchicine or MithA attenuated histopathological damage in lung of hamsters infected with SARS-CoV-2. Representative images of H&E staining in lung of hamsters infected with SARS-CoV-2 at 3 dpi (**E**). Bronchial epithelial cell necrosis and pyknosis (black arrow), edema, loose arrangement of muscle fibers, and massive lymphocyte infiltration (blue arrow), extensive hemorrhage (green arrow), and alveolar wall thickening (yellow arrow). Parts on lower side are high-power images of black and blue arrows, respectively. Scale bar: 40 μm. Representative images of Masson staining in lung of hamsters infected with SARS-CoV-2 at 3 dpi (**F**). Scale bar: 40 μm. Summary of lung lesion scoring in different groups at 3 dpi (n=5 each group) (**G**). Error bars show means ± SD. **p<0.01 (Student's t-test). Quantitative analysis of fibrotic area in lung tissues (**H**). Error bars show means ± SD. **p<0.01 (Student's t-test). Ctrl, Control. Veh, Vehicle. Col, Colchicine. nCoV-2, SARS-CoV-2.

The online version of this article includes the following source data and figure supplement(s) for figure 6:

**Source data 1.** Original file for determination of viral load in *Figure 6C*.

**Source data 2.** Original file for determination of viral load in *Figure 6D*.

**Source data 3.** Original file for lesion scores in *Figure 6G*.

**Source data 4.** Original file for quantitative analysis of fibrotic area in *Figure 6H*.

**Figure supplement 1.** Schematic representation of experimental design for SARS-CoV-2 infection in Syrian hamsters.

**Figure supplement 2.** Colchicine and MithA blocked replication of SARS-CoV-2 by inhibiting ACE2 expression in both lung and trachea of hamsters.

**Figure supplement 2—source data 1.** Original file for quantification of ACE2 fluorescence intensity in *Figure 6—figure supplement 2A, B*.

**Figure supplement 2—source data 2.** Original file for quantification of SARS-CoV-2-N fluorescence intensity in *Figure 6—figure supplement 2C, D*.

**Figure supplement 2—source data 3.** Original file for quantification of p-SP1 fluorescence intensity in *Figure 6—figure supplement 2E, F*.

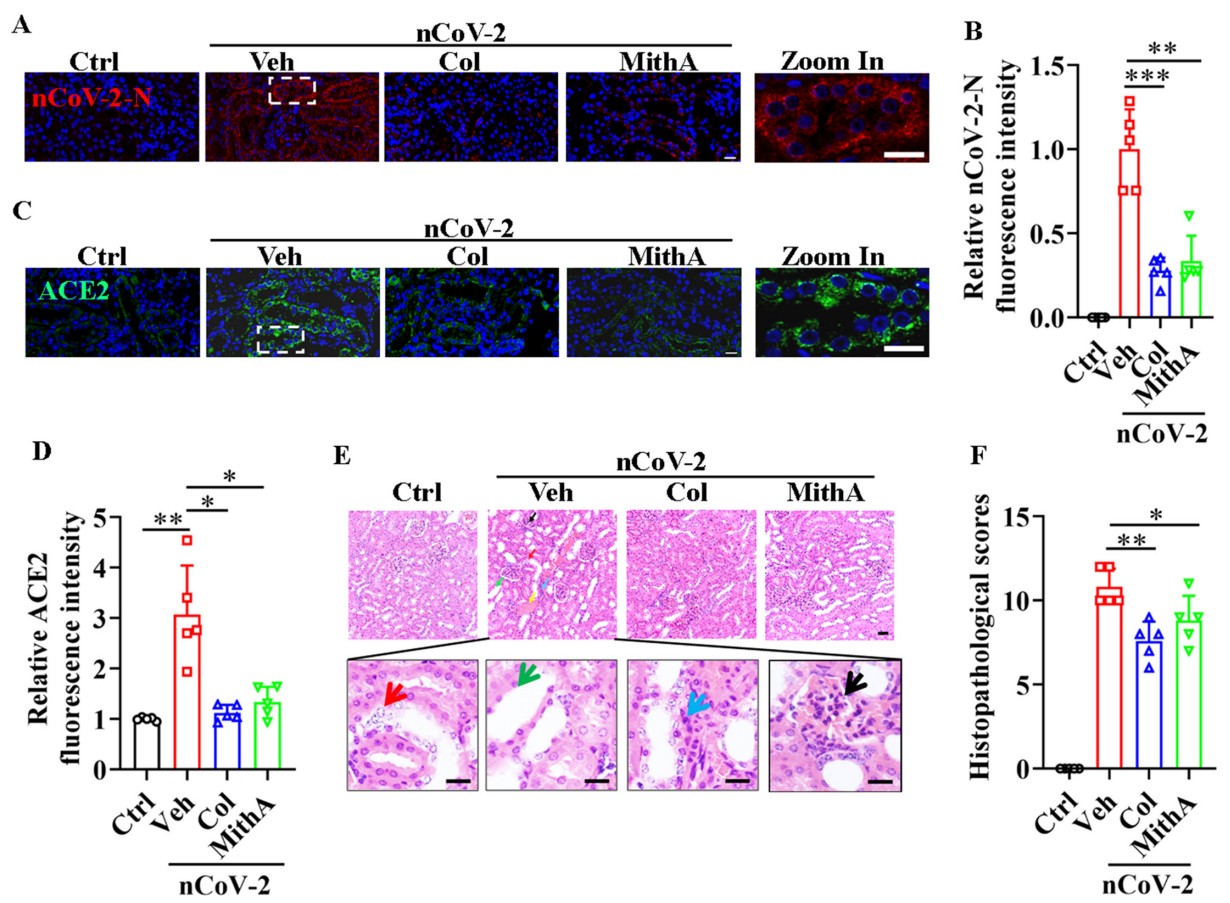

**Figure 7.** Inhibition of SP1 inhibited SARS-CoV-2 replication and reduced kidney pathology in Syrian hamsters. (**A** and **B**) Treatment with colchicine or MithA inhibited SARS-CoV-2 replication in kidney of hamsters. Representative images of immunofluorescence staining of SARS-CoV-2-N in kidney of hamsters (**A**). Parts on right side are high-power images. Scale bar: 20 µm. Quantification of SARS-CoV-2-N fluorescence intensity in kidney of hamsters (**B**). Error bars show means ± SD. **p<0.01, ***p<0.001 (Student's *t*-test). (**C** and **D**) Treatment with colchicine or MithA inhibited ACE2 expression in kidney of hamsters infected with SARS-CoV-2. Representative images of immunofluorescence staining of ACE2 in kidney of hamsters (**C**). Parts on right side are high-power images. Scale bar: 20 µm. Quantification of ACE2 fluorescence intensity in kidney of hamsters (**D**). Error bars show means ± SD. *p<0.05, **p<0.01 (Student's *t*-test). (**E** and **F**) Supplementation with colchicine or MithA attenuated histopathological damage in kidney of hamsters infected with SARS-CoV-2. Representative images of H&E staining in kidney of hamsters infected with SARS-CoV-2 at 3 dpi (**E**). Histopathology of kidney showed renal interstitial vascular congestion (black arrow), renal tubular epithelial cell nuclear pyknosis (red arrow), brush border disappearance (green arrow), renal interstitial inflammatory cell infiltration (blue arrow), and glomerular atrophy (yellow arrow). Parts on lower side are high-power images of red, green, and blue arrows, respectively. Scale bar: 40 µm. Summary of kidney lesion scoring in different groups at 3 dpi (n=5 each group) (**F**). Error bars show means ± SD. *p<0.05, **p<0.01 (Student's *t*-test). Ctrl, Control. Veh, Vehicle. Col, Colchicine. nCoV-2, SARS-CoV-2.

The online version of this article includes the following source data for figure 7:

**Source data 1.** Original file for quantification of SARS-CoV-2-N fluorescence intensity in *Figure 7B*.

**Source data 2.** Original file for quantification of ACE2 fluorescence intensity in *Figure 7D*.

**Source data 3.** Original file for lesion scores in *Figure 7F*.

negative regulatory effects, respectively. Notably, our findings indicated that the PI3K/AKT signaling cascade, activated following SARS-CoV-2 infection (*Klann et al., 2020*), served as a crucial regulatory node in the induction of ACE2 expression by enhancing and reducing the transcriptional activities of SP1 and HNF4α, respectively.

Our findings also showed that inhibition of SP1 activity, leading to the down-regulation of ACE2 expression, effectively mitigated SARS-CoV-2 replication in the lung and trachea of Syrian hamsters. Importantly, this decrease in viral replication was associated with a marked reduction in lung pathology. In addition to lung-associated issues, acute kidney injury has also been identified as an extrapulmonary manifestation of severe COVID-19 (*Legrand et al., 2021*), although the direct causal association

between SARS-CoV-2 infection and acute kidney injury remains controversial (*Legrand et al., 2021*; *Smith and Akilesh, 2021*). Recent evidence has indicated that SARS-CoV-2 can directly infect kidney cells in human-induced pluripotent stem cell-derived kidney organoids (*Jansen et al., 2022*). Consistent with these findings, we found that colchicine treatment effectively suppressed ACE2 expression in the proximal kidney tubule, which was associated with diminished SARS-CoV-2-induced kidney injury in Syrian hamsters. These results support ACE2 as a promising therapeutic target against COVID-19 (*Monteil et al., 2020*). Notably, proxalutamide, a potent androgen receptor antagonist, has been shown to suppress androgen-induced ACE2 expression (*Qiao et al., 2021*). Moreover, RIPK1 inhibition has been found to reduce viral loads in SARS-CoV-2-infected human lung organoids, accompanied by a down-regulation in the transcriptional induction of ACE2 (*Xu et al., 2021a*). A recent study also demonstrated that inhibition of FXR activity via UDCA can lead to the down-regulation of ACE2 expression, consequently reducing SARS-CoV-2 infection in vitro, in vivo, and ex vivo (*Brevini et al., 2023*).

In further examination of ACE2 expression, an unexpected observation emerged. Notably, inhibition of SP1 by its inhibitor MithA induced nuclear accumulation of HNF4α, whereas inhibition of HNF4α by its antagonist BI6015 increased phosphorylation levels of SP1 in the nucleus. Further exploration is warranted to elucidate the mechanisms underlying the antagonistic interactions between the two transcription factors. Although the established role of MithA is to disrupt the binding of SP1 to its consensus site (*Lee et al., 2011*), it is also capable of inducing proteasome-dependent SP1 degradation (*Choi et al., 2014*; *Lee et al., 2012*). This may explain why MithA treatment significantly suppressed the phosphorylation of SP1 in our study. Similarly, BI6015, known to repress the DNA-binding activity HNF4α (*Kiselyuk et al., 2012*), can also diminish HNF4α expression. This may provide a ready explanation of our observation that BI6015 could reduce nuclear accumulation of HNF4α induced by colchicine. Our results demonstrated that non-phosphorylated SP1 interacted with HNF4α and potentially inhibited the phosphorylation of SP1. Notably, upon reduction of HNF4α protein levels by BI6015, SP1 was released and subsequently phosphorylated by AKT, resulting in the binding of phosphorylated SP1 to the GC box and subsequent up-regulation of ACE2 expression. Conversely, upon reduction of SP1 protein levels by MithA, HNF4α was released, which inhibited ACE2 expression by binding to the HNF4α-specific binding motif. These observations enhance our understanding of the counteracting transcriptional activities between SP1 and HNF4α. However, additional research is needed to elucidate these mechanisms, particularly in the context of current findings.

This study has several limitations. Firstly, although SP1 was identified as a pivotal transcription factor in modulating ACE2 expression via the action of colchicine and MithA, neither of these compounds currently qualify as a candidate for the treatment of COVID-19. Primarily used to treat several types of cancer, including testicular cancer (*Kennedy and Torkelson, 1995*) and myeloid leukemia (*Dutcher et al., 1997*), the clinical application of MithA is substantially limited by its range of adverse effects, including liver, kidney, gastrointestinal, and bone marrow toxicity (*Green and Donehower, 1984*; *Kennedy, 1970*; *Quarni et al., 2019*). Additionally, the efficacy of colchicine as a treatment for COVID-19 remains inconclusive. While some studies suggest benefits (*Chiu et al., 2021*; *Drosos et al., 2022*; *Elshafei et al., 2021*), others indicate negligible impact on mortality or disease progression (*RECOVERY Collaborative Group, 2021*; *Mikolajewska et al., 2021*). Secondly, although inhibition of SP1 phosphorylation in Syrian hamsters was shown to effectively reduce viral replication and inflammatory responses, these effects need to be further evaluated in other model animals, such as ferrets and non-human primates (*Shi et al., 2020*).

Increased ACE2 expression in respiratory and pulmonary tissues has been implicated in the exacerbation of COVID-19 symptom severity among elderly individuals (*Inde et al., 2021*). Importantly, the Omicron variant of SARS-CoV-2, the current dominant global strain, exhibits a spike protein with a six- to nine-fold enhanced binding affinity for ACE2 (*Yin et al., 2022*). Thus, our observations have the potential to inform the development of novel host-targeting strategies to combat COVID-19, particularly in elderly populations affected by the SARS-CoV-2 Omicron variant.

# Materials and methods

## Cell culture

Immortalized human alveolar epithelial cells (HPAEpiCs) were generated from human lung tissue type II pneumocytes purchased from the ScienCell Research Laboratory (San Diego, CA, USA) and maintained in RPMI 1640 medium (01-100-1ACS, Biological Industries, Israel) supplemented with 10% fetal bovine serum (FBS) (C04001−050X10, VivaCell, Shanghai, China) and 1% penicillin-streptomycin. A549 cells were maintained in Dulbecco's Modified Eagle Medium-Nutrient Mixture F12 (DMEM/F12) (C3130-0500, VivaCell) containing 10% FBS and 1% penicillin-streptomycin. All cells were cultured in 5% $CO_2$, 95% air incubator at 37 °C.

The HPAEpiCs and A549 cells were treated with colchicine (C804812, Macklin, Shanghai, China), mithramycin A (MithA, A600668, Sangon Biotech, Shanghai, China), BI6015 (HY-108469, Med Chem Express, Shanghai, China), LY294002 (HY-10108, Med Chem Express), or wortmannin (HY-10197, Med Chem Express). Dimethyl sulfoxide (DMSO) was used as a control. Cells were infected with SARS-CoV-2 at a multiplicity of infection (MOI) of 1 for 1 hr.

## SARS-CoV-2

The SARS-CoV-2 strain (accession number: NMDCN0000HUI) was provided by the Guangdong Provincial Center for Disease Control and Prevention (Guangzhou, China). The virus was propagated in African green monkey kidney epithelial cells (Vero-E6) (ATCC, No. 1586) and titrated. All infection experiments were performed in a Biosafety Level-3 (BLS-3) Laboratory.

## Half-maximal effect concentration ($EC_{50}$)

The HPAEpiCs were seeded at a density of $1.6 \times 10^4$ cells/well in 48-well plates and grown overnight at 37 °C. The cells were infected with SARS-CoV-2 at an MOI of 1 and the test compounds were added to the wells at different concentrations. After 1 hr of incubation at 37 °C, the virus-drug mixture was removed and washed three times with phosphate-buffered saline (PBS) to eliminate free virus, before being replenished with fresh medium containing the compounds. After 48 hr, the supernatants were collected to extract viral RNA for RT-qPCR analysis. The $EC_{50}$ values were calculated using a dose-response model in GraphPad Prism v8.0 (GraphPad Software Inc, La Jolla, CA, USA).

## Cellular antiviral activity assay

The HPAEpiCs were seeded at a density of $4 \times 10^5$ cells/well in 24-well plates and grown overnight at 37 °C. Following preincubation with the test compounds for 2 hr, the cells were infected with SARS-CoV-2 at an MOI of 1. After 1 hr of incubation at 37 °C, the virus-drug mixtures were replaced with fresh medium containing compounds. After 24 hr, the cells were collected to extract total RNA and total cell protein. Viral RNA was quantified using a Thunderbird Probe One-step RT-qPCR Kit (QRZ-101, Toyobo, Shanghai, China). The TaqMan primers used for SARS-CoV-2 were 5'-GGG GAA CTT CTC CTG CTA GAA T-3' and 5'-CAG ACA TTT TGC TCT CAA GCT G-3' with SARS-CoV-2 probe FAM-TTG CTG CTG CTT GAC AGA TT-TAMRA-3'.

## RT-qPCR

Total RNA was extracted from trachea and lung tissues using RNAiso Plus (Takara, Dalian, China). Total RNA was extracted from cells using TRIzol Reagent (R1100, Solarbio, Shanghai, China), and reverse-transcribed into cDNA using a FastKing RT Kit (KR116, TIANGEN, Beijing, China). RT-qPCR analysis was performed using SuperReal PreMix Plus (SYBR Green) (FP205, TIANGEN) on a Roche LightCycler 480 System (Roche Applied Science, Mannheim, Germany). The primers used for RT-qPCR were: ACE2 (Forward 5'-GGG ATC AGA GAT CGG AAG AAG AAA-3'; Reverse 5'-AGG AGG TCT GAA CAT CAT CAG TG-3'); ACTB (Forward 5'-CCC TGG ACT TCG AGC AAG AG-3'; Reverse 5'-ACT CCA TGC CCA GGA AGG AA-3'). Beta-actin was used to calculate relative expression normalized to an internal control.

## 45-Pathway Reporter Array

Cignal Finder 45-Pathway Reporter Arrays (CCA-901, Qiagen, Hilden, Germany) were used according to the manufacturer's instructions to identify potential pathways regulated by SARS-CoV-2. Briefly,

HPAEpiCs were reverse transfected with firefly luciferase reporter constructs containing response elements for the indicated pathways, and with control *Renilla* luciferase constructs for 24 hr. Cells were then pretreated with colchicine for 2 hr and incubated with SARS-CoV-2 for 24 hr. Luciferase activities of the cells were then measured using a dual-luciferase reporter assay system (E1910, Promega) on a fluorescent microplate reader (Molecular Devices Inc). Reporter luciferase activity was normalized to *Renilla* luciferase activity for each sample. All experiments were performed with three biological replicates.

## Immunofluorescence

Cells were fixed with 4% paraformaldehyde (PFA) for 10 min at room temperature. Paraformaldehyde-fixed, paraffin-embedded tissue samples was cut into 4 μm slices and adhered to frosted glass slides. After washing with PBS and treating with PBS containing 0.1% Triton X-100 for 15 min, the sections and cells were permeabilized and blocked with 0.1% Tween-20 in PBS (PBST) containing 5% FBS for 90 min at room temperature. The cells were immunostained with anti-ACE2 (ab15348, 1:500, Abcam), anti-SP1 (T453) (ab59257, 1:500, Abcam), or anti-HNF4α antibodies (3113, 1:1 000, Cell Signaling Technology) overnight at 4 °C. The tissue sections were immunostained with anti-ACE2 (GB11267, 1:200**,** Servicebio, Wuhan, China) or anti-SARS-CoV-2-N antibodies (40143-MM05, 1:500, SinoBiological, Beijing, China) overnight at 4 °C. After washing three times with 0.1% Tween-20 in PBS (PBST), the cells were incubated with Alexa Fluor 594 anti-rabbit IgG (H+L) (A-21207, 1:200, ThermoFisher Scientific), Cy3 conjugated goat anti-mouse IgG (H+L) (GB21301, 1:300**,** Servicebio), or Alexa Fluor 488-conjugated goat anti-rabbit IgG (H+L) (GB25303, 1:500**,** Servicebio) at room temperature for 1 hr. After staining with primary antibodies, nuclei were counterstained with 4′,6-diamidino-2-phenylindole (DAPI). Images were acquired using a Zeiss Axioskop 2 plus fluorescence microscope (Carl Zeiss, Jena, Germany).

## Luciferase reporter assay

The HPAEpiCs were co-transfected at a density of $3 \times 10^3$ with pRL-SV40 vector and phACE2-promoter-TA-luc (D2488, Beyotime) using Lipofectamine 3000 reagent (L3000015, Invitrogen). After 48 hr of transfection, luciferase activity was measured using the dual-luciferase reporter assay system (E1910, Promega, Shanghai, China) on a fluorescent microplate reader (Molecular Devices Inc, Sunnyvale, CA, USA). The ratio of firefly luciferase to *Renilla* luciferase was calculated for each experiment and averaged from three replicates. All experiments were performed with three biological replicates.

## Transcription factor binding motif enrichment

The ACE2 promoter sequence (1 500 bp upstream of TSS) was extracted from the NCBI database. The DNA-binding motifs of transcription factors estrogen receptor GATA6, HNF4α, NF-κB, and SP1 were obtained from the JASPAR CORE database (genereg.net). MEME Suite (meme-suite.org) was used to determine the enrichment of the transcription factor motifs and binding sites in the ACE2 promoter sequence (*Bailey et al., 2015*). FIMO analysis was performed using stringent criteria, including $p < 1E-4$ and a maximum of two mismatched residues.

## ChIP-qPCR

ChIP was performed as described previously (*Tao et al., 2016*) using a ChIP assay kit (P2078, Beyotime) following the manufacturer's directions. After crosslinking with formaldehyde, the chromatin solutions were sonicated and incubated with anti-SP1 (9389, 1:100, Cell Signaling Technology) and anti-HNF4α (ab181604, 1:100, Abcam) antibodies and control IgG, then rotated overnight at 4 °C, respectively. After purification using a DNA purification kit (BioTeke Corp.), the immunoprecipitated DNA was detected for PCR analysis. All ChIP-qPCR experiments were performed with three biological replicates.

## RNA interference for cells

All chemically synthesized siRNAs were obtained from the Gene-Pharma Corporation (Shanghai, China). To silence HNF4α, SP1, or AKT expression by siRNA, HPAEpiCs were seeded at a density of $5 \times 10^5$ cells/well in six-well plates containing complete culture medium. After 24 hr, the cells were transiently transfected with 100 nM of siRNAs using Lipofectamine 3000 (L3000015, Invitrogen, Beijing,

China). Gene silencing efficiency was confirmed by RT-qPCR at 48 hr post-transfection. The following siRNAs were used (sense strand sequences): HNF4α, 5'-GUC AUC GUU GCC AAC ACA AUG-3'; SP1, 5'-CUC CAA GGC CUG GCU AAU AAU-3'; AKT, 5'-CGC GUG ACC AUG AAC GAG UUU-3'; negative control, 5'-UUC UCC GAA CGU GUC ACG UUU-3'.

## Co-immunoprecipitation (co-IP)

For the co-IP experiments, HPAEpiCs were lysed on ice for 30 min in cell lysis buffer (P0013, Beyotime, Shanghai, China). After centrifugation at 12,000 rpm for 30 min at 4 °C, the supernatant was collected and incubated with anti-HNF4α antibodies (ab181604, 1:70, Abcam, Shanghai, China) overnight. After 4 hr of incubation with Protein A Agarose (20333, Thermo Scientific, Shanghai, China) at 4 °C, the complexes were washed three times. Immunoblotting was performed after elution.

## Western blotting

For measurement of protein expression, HPAEpiCs were re-suspended in RIPA buffer (R0278, Sigma-Aldrich, Shanghai, China) on ice for 1 hr. The protein samples were separated using sodium dodecyl-sulfate polyacrylamide gel electrophoresis (SDS-PAGE), then transferred to polyvinylidene fluoride (PVDF) membranes. After blocking with 5% bovine serum albumin (BSA) in PBS-T buffer containing 0.05% Tween-20, the membranes were incubated at 4 °C overnight with primary antibodies, including anti-ACE2 (ab108252, 1:1 000, Abcam), anti-HNF4α (ab181604, 1:1 000, Abcam), anti-phospho-AKT (Thr308) (13038, 1:1 000, Cell Signaling Technology, Shanghai, China), anti-phospho-AKT (Ser473) (4060, 1:1 000, Cell Signaling Technology), anti-pan-AKT (4691, 1:1 000, Cell Signaling Technology), anti-phospho-SP1 (T453) (ab59257, 1:1 000, Abcam), and anti-Actin antibodies (sc-47778, 1:5 000, Santa Cruz-Biotechnology, Shanghai, China). After washing with PBS-T, the membranes were incubated for 2 hr at room temperature with horseradish peroxidase (HRP)-conjugated anti-mouse (7076, 1:2 000, Cell Signaling Technology) or anti-rabbit IgG secondary antibodies (7074, 1:2 000, Cell Signaling Technology). Protein bands were detected using ECL (RPN2232, GE Healthcare, Little Chalfont, UK) on an Amersham Imager 600. Subsequent image analysis was performed using ImageJ software. All experiments were performed with three biological replicates.

## Animal experiments and in vivo procedures

Male Syrian hamsters (*Mesocricetus auratus*) were purchased from Beijing Vital River Laboratory Animal Technology Co., Ltd. (China) (85–100 g, 5 weeks old). All animals used were chosen randomly. Colchicine (C804812, Macklin, Shanghai, China) and MithA (A600668, Sangon Biotech, Shanghai, China) were dissolved in 1% (v/v) DMSO and 99% saline. The hamsters were anesthetized by inhalation of isoflurane and infected with $10^4$ $TCID_{50}$ of SARS-CoV-2 by intranasal instillation. After 1 hr, 0.2 mg/kg of either colchicine or MithA was given intraperitoneally once daily. Trachea, lung, and kidney tissues were collected 3 dpi.

## Hematoxylin and eosin (H&E) staining and histopathology scores

Lung, trachea, and kidney samples were collected from the Syrian hamsters, then fixed with 4% para-formaldehyde, embedded in paraffin, and sectioned (4 μm thick). The tissue sections were then stained with H&E for histopathological examination. After staining, a four-point scoring system was applied to assess the severity of pathology in tissues, evaluated by a trained pathologist (L.-Q.W.) blind to group identity. Scoring was graded from 0, indicating no pathological change, to 1–4, indicating increasing severity. Lung histopathological scores were assessed based on alveolar wall thickening, edema, hemorrhage, and inflammatory cell infiltration. Kidney histopathological scores were assessed based on cellular degeneration, necrosis, hemorrhage, inflammatory cell infiltration, and congestion. Histopathological scores represented the sum of the injury subtype scores for each condition on a 0–20 scale.

## Statistical analysis

Differences in gene expression, mRNA and protein levels, viral RNA, Luciferase reporter assay, ChIP-qPCR assay, and fluorescence intensity were assessed by Student's *t*-test. Data were analyzed using GraphPad Prism v8 (GraphPad Software Inc, La Jolla, CA, USA).

## Acknowledgements

We are grateful to Ms. Christine Watts for her critical reading of this manuscript. We thank Dr. Changwen Ke (Guangdong Provincial Center for Disease Control and Prevention) for providing the SARS-CoV-2 strain. We appreciate the support from the Kunming National High-level Biosafety Research Center for Non-human Primates, Kunming Institute of Zoology, Chinese Academy of Sciences. This work was supported in part by grants from the National Key Research and Development Program of China (2021YFC2301303, 2022YFC2303700), Yunnan Key Research and Development Program (202103AC100005, 202103AQ100001, 202102AA310055), and Major Science and Technology Project in Yunnan Province of China (202001BB050001).

## Additional information

### Funding

| Funder | Grant reference number | Author |
|---|---|---|
| National Key Research and Development Program of China | 2021YFC2301303 | Yong-Tang Zheng |
| National Key Research and Development Program of China | 2022YFC2303700 | Yong-Tang Zheng |
| Yunnan Key Research and Development Program | 202103AC100005 | Yong-Tang Zheng |
| Yunnan Key Research and Development Program | 202103AQ100001 | Yong-Tang Zheng |
| Yunnan Key Research and Development Program | 202102AA310055 | Yong-Tang Zheng |
| Major Science and Technology Projects in Yunnan Province | 202001BB050001 | Cheng-Gang Zou |

The funders had no role in study design, data collection and interpretation, or the decision to submit the work for publication.

### Author contributions

Hui Han, Data curation, Formal analysis, Visualization, Methodology, Writing – original draft; Rong-Hua Luo, Xin-Yan Long, Li-Qiong Wang, Qian Zhu, Xin-Yue Tang, Rui Zhu, Methodology; Yi-Cheng Ma, Data curation, Formal analysis, Visualization, Writing – original draft; Yong-Tang Zheng, Funding acquisition; Cheng-Gang Zou, Conceptualization, Funding acquisition, Investigation, Methodology, Writing – original draft

### Author ORCIDs

Cheng-Gang Zou ⓘ https://orcid.org/0000-0001-5519-4402

### Ethics

In this study, all animal experimental procedures were approved by the Institutional Committee for Animal Care and Biosafety at Kunming Institute of Zoology, Chinese Academy of Sciences (Permit Number: IACUC-RE-2021-10-002).

### Decision letter and Author response

Decision letter https://doi.org/10.7554/eLife.85985.sa1
Author response https://doi.org/10.7554/eLife.85985.sa2

# Additional files

## Supplementary files
• MDAR checklist

## Data availability
All data generated or analysed during this study are included in the manuscript and supporting files; source data files have been provided for *Figures 1–7*.

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
