## [Editor Report]

This is a valuable report that describes that ACE2 expression is upregulated by SARS-CoV-2 infection via activation of transcription factor Sp1 and inhibition of HNF4α through the PI3K/AKT pathway. Inhibition of Sp1 reduces SARS-CoV-2 infection in vitro and in an animal model. This work is solid and will be of interest to those interested in ACE2 biology and its impact in COVID-19.

---

## [Decision Letter]

**Decision letter after peer review:**

Thank you for submitting your article "Transcription regulation of SARS-CoV-2 receptor ACE2 by Sp1: a potential therapeutic target" for consideration by *eLife*. Your article has been reviewed by 2 peer reviewers, and the evaluation has been overseen by a Reviewing Editor and Jos van der Meer as the Senior Editor.

*Reviewer #1 (Recommendations for the authors):*

Language: I suggest moderate changes to the English language.

Line 103: Colchicine is used for more than just acute gout and familial Mediterranean fever. More correct would be to write "used for the treatment of (auto-)inflammatory disorders such as acute gout or familial Mediterranean fever"

Figure 2C: For the experiment carried out for Figure 2C, we should know whether cells not infected by SARS-CoV-2 but that are merely in the vicinity of infected cells also express increased p-Sp1. If only cells with active infection of SARS-CoV-2 express increased p-Sp1, this would not add to the spread of the infection (as these cells are already infected). Please add similar experiments in which SARS-CoV-2 virus is also stained (be it by RNAscope or by protein staining).

Discussion: The discussion lacks a "limitations" section, please add this.

*Reviewer #2 (Recommendations for the authors):*

The authors sought to identify the transcription factors that regulate ACE2 expression. The phenotype of Sp1 and HNF4α was mild in the study, and direct evidence that shows PI3K/AKT inhibitors work through Sp1 was lacking.

Sp1 inhibitor was used to demonstrate the antiviral activity in vivo. It would be more convincing to have genetic data besides pharmacological inhibitors due to specificity concerns.

---

## [Author Response]

Reviewer #1 (Recommendations for the authors):Language: I suggest moderate changes to the English language.Line 103: Colchicine is used for more than just acute gout and familial Mediterranean fever. More correct would be to write "used for the treatment of (auto-)inflammatory disorders such as acute gout or familial Mediterranean fever"

We really appreciate this reviewer’s preciseness. Amended.

Figure 2C: For the experiment carried out for Figure 2C, we should know whether cells not infected by SARS-CoV-2 but that are merely in the vicinity of infected cells also express increased p-Sp1. If only cells with active infection of SARS-CoV-2 express increased p-Sp1, this would not add to the spread of the infection (as these cells are already infected). Please add similar experiments in which SARS-CoV-2 virus is also stained (be it by RNAscope or by protein staining).

We appreciate the concerns of the reviewer. In response to the reviewer’s suggestion, we detected the expression of p-SP1 and SARS-CoV-2 N protein in HPAEpiCs by immunofluorescence. We found that p-SP1 levels were increased only in cells infected with SARS-CoV-2, but not in non-infected cells (Author response image 1). One may speculate that SARS-CoV-2 cell entry induces the phosphorylation of SP1 to up-regulate the expression of ACE2, leading to more infectious viruses in infected cells. The virus uses the cell to make more viruses. Thus, adjacent cells are at increased risk of virus infection. By contrast, inhibition of ACE2 expression by inhibiting the phosphorylation of SP1 reduces the spread of the virus between cells.

**Author response image 1. sa2fig1:** Representative images of immunofluorescence staining for p-SP1 and SARS-CoV-2. SARS-CoV-2 significantly increased the levels of p-SP1 in infected HPAEpiCs, whereas the p-SP1 levels were unchanged in non-infected cells. Veh, Vehicle. Col, Colchicine. nCoV-2, SARS-CoV-2. Scale bar: 10 μm.

Discussion: The discussion lacks a "limitations" section, please add this.

Amended. In response to the reviewer’s comments, we added the sentences “This study has several limitations. Firstly, although SP1 was identified as a pivotal transcription factor in modulating ACE2 expression via the action of colchicine and MithA, neither of these compounds currently qualify as a candidate for the treatment of COVID-19. Primarily used to treat several types of cancer, including testicular cancer…” in Discussion of revised manuscript (Lines 329-342).

Reviewer #2 (Recommendations for the authors):The authors sought to identify the transcription factors that regulate ACE2 expression. The phenotype of Sp1 and HNF4α was mild in the study, and direct evidence that shows PI3K/AKT inhibitors work through Sp1 was lacking.

We appreciate the concerns of the reviewer. In response to the reviewer’s suggestion, we tested the effect of SP1 knockdown or overexpression on ACE2 expression in the presence or absence of PI3K inhibitors in HPAEpiCs. We found that ACE2 expression was decreased in both PI3K inhibitor-treated cells and HPAEpiCs subjected to SP1 knockdown (Author response image 2). In contrast, SP1 overexpression significantly increased the expression of ACE2 in HPAEpiCs (Author response image 2). In addition, SP1 knockdown did not affect the expression of ACE2 in PI3K inhibitor-treated cells (Author response image 2), whereas SP1 overexpression rescued the expression of ACE2 in PI3K inhibitor-treated cells (Author response image 2). Similar results were observed by immunofluorescent staining of ACE2 in HPAEpiCs (Author response image 2). These results indicate that the PI3K/AKT pathway regulates the expression of ACE2 through SP1.

**Author response image 2. sa2fig2:** The PI3K/AKT pathway regulates the expression of ACE2 through SP1. (A and B) HPAEpiCs were treated with LY294002 (30 μM), wortmannin (50 nM), siSP1, siSP1 + LY294002 or siSP1 + wortmannin, respectively. Either knockdown of SP1 or inhibition of PI3K significantly suppressed the levels of ACE2 in HPAEpiCs. The blot is typical of three independent experiments (A). Quantification the ratio of ACE2 to Actin (B). These results are means ± SD of three independent experiments. ns, not significant, **P* < 0.05. (C and D) SP1 overexpression rescued the expression of ACE2 in PI3K inhibitor-treated cells. The blot is typical of three independent experiments (C). Quantification the ratio of ACE2 to Actin (D). These results are means ± SD of three independent experiments. **P* < 0.05. (E) Representative images of immunofluorescence staining for ACE2. Scale bar: 10 μm. (F) Quantification of ACE2 fluorescence intensity. These results are means ± SD of three independent experiments. **P* < 0.05, ***P* < 0.01, ****P* < 0.001, ns, not significant. NC, Negative Control. LY, LY294002. Wort, wortmannin.

Sp1 inhibitor was used to demonstrate the antiviral activity in vivo. It would be more convincing to have genetic data besides pharmacological inhibitors due to specificity concerns.

We appreciate the concerns of the reviewer. In response to the reviewer’s suggestion, we detected the p-SP1 levels in lung and trachea of hamsters infected with SARS-CoV-2 after colchicine and MithA treatment. SARS-CoV-2 increased the levels of p-SP1 in the lung and trachea of hamsters, while treatment of either colchicine or MithA reduced the levels of p-SP1 in the lung and trachea of hamsters infected with SARS-CoV-2. We have included these data in the revised manuscript (see revised Figure 6B in the section of Results of revised manuscript).

References

Brevini, T., Maes, M., Webb, G. J., John, B. V., Fuchs, C. D., Buescher, G., et al. (2022). FXR inhibition may protect from SARS-CoV-2 infection by reducing ACE2. *Nature*, 615(7950), 134-142.

Chen, J., Fan, J., Chen, Z., Zhang, M., Peng, H., Liu, J., et al. (2021). Nonmuscle myosin heavy chain IIA facilitates SARS-CoV-2 infection in human pulmonary cells. *Proceedings of the National Academy of Sciences of the United States of America, 118*(50), e2111011118.

Chiu, L., Lo, C. H., Shen, M., Chiu, N., Aggarwal, R., Lee, J., et al. (2021). Colchicine use in patients with COVID-19: A systematic review and meta-analysis. *PLoS One, 16*(12), e0261358.

Drosos, A. A., Pelechas, E., Drossou, V., & Voulgari, P. V. (2022). Colchicine against SARS-CoV-2 infection: what is the evidence? *Rheumatology and Therapy, 9*(2), 379-389.

Elshafei, M. N., El-Bardissy, A., Khalil, A., Danjuma, M., Mubasher, M., Abubeker, I. Y., et al. (2021). Colchicine use might be associated with lower mortality in COVID-19 patients: A meta-analysis. *European Journal of Clinical Investigation, 51*(9), e13645.

Group, R. C. (2021). Colchicine in patients admitted to hospital with COVID-19 (RECOVERY): a randomised, controlled, open-label, platform trial. *The Lancet Respiratory Medicine, 9*(12), 1419-1426.

Mikolajewska, A., Fischer, A. L., Piechotta, V., Mueller, A., Metzendorf, M. I., Becker, M., et al. (2021). Colchicine for the treatment of COVID-19. *Cochrane Database of Systematic Reviews, 10*(10), Cd015045.

Qiao, Y. Y., Wang, X. M., Mannan, R., Pitchiaya, S., Zhang, Y. P., Wotring, J. W., et al. (2021). Targeting transcriptional regulation of SARS-CoV-2 entry factors ACE2 and TMPRSS2. *Proceedings of the National Academy of Sciences of the United States of America*, 118(1), e2021450118.

Samelson, A. J., Tran, Q. D., Robinot, R., Carrau, L., Rezelj, V. V., Kain, A. M., et al. (2022). BRD2 inhibition blocks SARS-CoV-2 infection by reducing transcription of the host cell receptor ACE2. *Nature Cell Biology, 24*(1), 24-34.

Samuel, R. M., Majd, H., Richter, M. N., Ghazizadeh, Z., Zekavat, S. M., Navickas, A., et al. (2020). Androgen signaling regulates SARS-CoV-2 receptor levels and is associated with severe COVID-19 symptoms in men. *Cell Stem Cell*, 27(6), 876-889.e812.

Shen, X. R., Geng, R., Li, Q., Chen, Y., Li, S. F., Wang, Q., et al. (2022). ACE2-independent infection of T lymphocytes by SARS-CoV-2. *Signal Transduction and Targeted Therapy 7*(1), 83.

Yeung, M. L., Teng, J. L. L., Jia, L., Zhang, C., Huang, C., Cai, J., et al. (2021). Soluble ACE2-mediated cell entry of SARS-CoV-2 via interaction with proteins related to the renin-angiotensin system. *Cell, 184*(8), 2212-2228.e2212.